# Stylohyoid Chain Syndrome (Eagle Syndrome) in Conjunction with Carotid Artery Dissection: A Case Report of Surgical Treatment

**DOI:** 10.3390/diseases12010024

**Published:** 2024-01-12

**Authors:** Jakub Bargiel, Michał Gontarz, Krzysztof Gąsiorowski, Tomasz Marecik, Paweł Wrona, Jan Zapała, Grażyna Wyszyńska-Pawelec

**Affiliations:** Department of Cranio-Maxillofacial Surgery, Jagiellonian University Medical College, 30-688 Cracow, Poland; michal.gontarz@uj.edu.pl (M.G.); krzysztof.gasiorowski@uj.edu.pl (K.G.); tomasz.marecik@uj.edu.pl (T.M.); pwrona@su.krakow.pl (P.W.); grazyna.wyszynska-pawelec@uj.edu.pl (G.W.-P.)

**Keywords:** eagle syndrome, stylalgia, carotid artery syndrome, carotid artery dissection, CAD, ischemic stroke, VSCS

## Abstract

(1) Background: “Eagle Syndrome”, also known as “stylalgia” or a “stylohyoid chain anomaly”, typically manifests with ipsilateral orofacial pain, a foreign body sensation in the throat, and ear-related symptoms. Despite these common presentations, its potential association with carotid artery dissection is not widely acknowledged. (2) Methods: This article presents an extremely rare case of a patient diagnosed with an ischemic stroke in the left hemisphere, followed by the dissection of the left internal carotid artery, initially with an unidentified cause. Subsequent examinations revealed elongated left styloid processes directly compressing the dissected artery. (3) Results: After initial treatment involving pharmacological and mechanical thrombectomy, styloidectomy restored blood flow to the internal carotid artery. The patient remained symptom-free during a 12-month follow-up. (4) Conclusions: This case emphasizes the importance of considering anatomical variations within the stylohyoid chain when assessing young individuals with neurological symptoms. Furthermore, it underscores the potential benefits of early surgical intervention in reducing the morbidity and mortality associated with this condition.

## 1. Introduction

The styloid process (SP) is a bony projection situated anteriorly to the stylomastoid foramen, measuring up to 30 mm, with its tip usually lying between the internal and external carotid artery. The elongation of the styloid process attributed to the ossification of the stylohyoid ligament may lead to various clinical symptoms widely recognized as Eagle’s Syndrome (ES) [1]. Eagle differentiated between two forms of the syndrome: the classic form, characterized by typical symptoms of orofacial pain (stylalgia) following tonsillectomy, and the carotid artery syndrome (CAS) [2]. Elongated styloid processes are relatively rare, with an incidence of between 3.3% and 35% in the general population. It is widely recognized that approximately 4% of individuals with this condition will experience symptoms [3,4].

Stylalgia occurs due to the irritation of cranial nerves (IX, X), sympathetic trunks, soft tissues within the parapharyngeal space, and the lateral wall of the pharynx. Persistent, one-sided orofacial and neck pain, a feeling of a foreign body in the throat, difficulty swallowing, tinnitus, and ear pain (otalgia) are common symptoms. It is typically diagnosed through a careful differential diagnosis and is no longer associated with post-tonsillectomy scarification.

CAS is thought to be linked to mechanical irritation of the nerve plexus, causing pain along the course of the internal or external carotid artery [5]. The alteration in the trajectory of the stylohyoid chain is considered a significant factor in this phenomenon. Direct contact with the internal carotid artery (ICA) may lead to hemodynamic disturbances and potential injury to the arterial wall, increasing the risk of dissection or vessel rupture [6,7].

Carotid Artery Dissection (CAD) is a complex clinical condition with multifactorial causes involving the tearing and separating of the innermost layer of the artery, known as the intima, due to the turbulent flow of incoming blood [8]. This tear can occur spontaneously or result from trauma. Risk factors for CAD encompass a family history of the disease, smoking, hypertension, genetic factors, and extensive physical exercise [9]. In addition to congenital and acquired factors, recent studies have identified structural abnormalities in the outer wall of the carotid associated with CAD [10]. CAD can manifest either extracranially or intracranially, potentially disrupting blood flow to specific brain regions and leading to a stroke [11]. An elongated styloid process (SP) is an exceedingly rare cause of spontaneous dissection [12]. The conjunction of Carotid Artery Syndrome and CAD is rare, with limited publications addressing this phenomenon. This case report describes the emergency decompression of a dissected carotid artery through cervical styloidectomy.

## 2. Case Report

A previously healthy 43-year-old man, with a history of nicotine use equivalent to 30 pack-years, was admitted to the Stroke Unit of Jagiellonian University Hospital in Cracow due to sudden dysarthria and right arm weakness experienced upon waking at 8:00 a.m. The patient also reported left-sided neck and chest pain extending to the left ear, occurring after strenuous physical activity helping a neighbor push a car the day before. On admission, the patient was fully conscious and oriented, displaying mild weakness in the right arm, moderate weakness in the right leg, and central facial paralysis. In the National Institutes of Health Stroke Scale (NIHSS), the patient scored 10 points.

A non-contrast CT revealed no fresh ischemic changes, intracranial bleeding, or expansive processes in the brain. CT angiography (CTA) disclosed occlusion of the left internal carotid artery (LICA). A non-contrast head MRI indicated a mismatch between FLAIR and DWI sequences, necessitating reperfusion therapy. The patient received 68 mg of Alteplase and was transferred to the Center for Interventional Treatment of Acute Stroke (CITO) at our Institution for mechanical thrombectomy.

Upon arrival at CITO, the patient displayed significant neurological deficits, including complete hemianopia, hemiplegia, right-sided paresthesia, and a positive Babinski sign on the right. The NIHSS score was tallied at 17, indicating a moderate level of stroke severity. A CT scan revealed an area of disrupted blood flow within the territory supplied by the occluded left middle cerebral artery (LMCA). Thrombectomy was conducted under analgosedation, successfully removing the thrombus material that had obstructed both the left internal carotid artery (LICA) and the M1 segment of the left middle cerebral artery (LMCA). Subsequent CTA demonstrated restored normal blood flow, and the decision to implant a stent in the LICA was postponed.

A thorough differential diagnosis was conducted to establish the stylohyoid origin of the dissection and concurrent stroke. This assessment systematically ruled out potential contributing factors, including inflammatory conditions, connective tissue disorders, metabolic irregularities, and coagulation abnormalities. The nasopharyngeal swab for SARS-CoV-2 infection yielded negative results.

During the CT angiography (CTA) evaluation of thrombolytic treatment, the radiologist identified a bony structure that reduced the diameter of the left internal carotid artery (LICA) from 14 to 7 mm. Three-dimensional reconstruction identified the styloid process, measuring 29 mm, compressing the vessel (Figure 1 and Figure 2). Vascular surgeons suggested LICA decompression without stenting. Consequently, the patient was qualified for styloidectomy and was transferred to the Cranio-Maxillo-Facial Department.

The surgery was performed under general anesthesia without the use of a neuromuscular blocking agent to preserve the function of the facial nerve. A curved incision was made in the skin crease situated 3 cm below the lower border and the angle of the mandible, with the head in a supine position and rotated to the contralateral site. The platysma flap was dissected at an identical level and elevated to reveal the anterior border of the sternocleidomastoideus muscle (SCM), external jugular vein, and submandibular gland. The gland was elevated with the Langenbeck hook to preserve and protect the marginal branch of the facial nerve. The SCM was retracted, and through gentle blunt dissection, the carotid sheath was exposed. After identification of the left internal carotid artery (LICA) at the CTA-identified level, it became evident that the styloid process, along with the stylohyoid ligament, directly compressed the dissected artery wall (Figure 3a). In order to alleviate compression on the left internal carotid artery (LICA) and eliminate the potential lateral movements of the bony fragment that posed a risk to the artery, the styloid process was cut using a diamond burr. Subsequently, it was delicately removed along with the calcified stylohyoid ligament (Figure 3b). The sharp edges of the remaining process were smoothed using the same method. Following the decompression of the LICA, the patient’s head was rotated to confirm the absence of contact in alternative positions. During this procedure, no vessel ligation was necessary. The surgical site was meticulously closed in layers and drained for a duration of two days.

## 3. Results

The surgical intervention successfully alleviated the compression on the left internal carotid artery induced by the styloid process, as confirmed by angioCT (Figure 4). The patient’s neurological condition showed significant improvement, with only minimal residual deficits observed during the follow-up. After three months, there was a residual minor facial muscle weakness on the right side, characterized by a flattened nasolabial fold and asymmetric smile. The patient scored 1 point on the NIHSS scale and 0 on the modified Rankin Scale (mRS). There was no need for stent grafting surgery.

Subsequently, a follow-up CTA performed after 12 months revealed the absence of carotid artery dissection. No residual neurological deficits were observed, and the patient reported the absence of a foreign body sensation, which he had not perceived as pathological before.

## 4. Discussion

In the late 19th century and the early 1900s, medical professionals began to recognize the potential clinical significance of abnormalities associated with the styloid process [13,14,15]. Eagle initially hypothesized a possible correlation between symptoms and the compression of sympathetic fibers of the carotid plexus by the styloid process. Unfortunately, his assumptions were based on only two clinical cases, rendering them inconclusive [16]. Given that carotid artery dissection (CAD) can be attributed to both elongated and regular styloid processes in conjunction with the stylohyoid ligament, we suggest the term “Vascular Stylohyoid Chain Syndrome (VSCS)”. This term encompasses a broader range of conditions wherein abnormalities in the styloid process and associated ligaments may contribute to vascular complications, such as CAD. This underscores the importance of thoroughly understanding these anatomical variations in clinical practice.

CAD is a significant contributor to stroke, occurring at a rate of 2.6 to 2.9 cases per 100,000 annually. It constitutes nearly 20% of strokes in young adults and approximately 2.5% in older patients [17]. However, the incidence of CAD may be underdiagnosed due to its manifestation with mild clinical symptoms. However, the incidence of CAD may be underestimated due to its manifestation with mild clinical symptoms. CAD often occurs spontaneously or after head and neck trauma and is more likely to occur during winter than in summer [18,19]. Risk factors for CAD encompass family history, extensive physical exercise, smoking, hypertension, Behcet’s disease, Marfan syndrome, Ehlers–Danlos syndrome, fibromuscular dysplasia, and other connective tissue disorders [9].

Only a few publications address the association between the styloid process and CAD. Some authors have reported detailed cases treated with stenting followed by styloidectomy [20,21]. Other authors have reported cases of CAD triggered by an elongated styloid process treated with antithrombotic therapy [22,23,24,25,26]. No documented correlation exists between the treatment of styloidectomy without stent grafting and Vascular Stylohyoid Chain Syndrome (VSCS).

Accurate diagnosis involves visualizing the styloid process and its relationship with neighboring structures. Diagnostic modalities range from conventional radiography and cone-beam computed tomography (CBCT) to computed tomography (CT) and magnetic resonance imaging (MRI). While radiographic techniques such as orthopantomogram may identify anomalies in the stylohyoid chain, CT angiography should be a gold standard. It provides three-dimensional capabilities that enable a thorough assessment of the relationship between the styloid process and blood vessels [27]. Unfortunately, there is no association between the length of the styloid process and the distance between the styloid process and the internal carotid artery with CAD [28]. Therefore, each case should be assessed independently.

In our study, physical exertion was the direct cause of the dissection of the left internal carotid artery (LICA), accompanied by compression on the artery wall by the stylohyoid chain. Thrombolytic treatment restored normal blood flow to the brain, leading to an improvement in neurological condition. A follow-up angioCT revealed a reduction in diameter with impaired blood flow in the left internal carotid artery (LICA), with the actual cause being the styloid process compressing its wall. Complete recovery was achieved after artery decompression by styloidectomy. To the best of our knowledge, this is the first publication regarding emergency decompression of dissected ICA by styloidectomy without stenting in VSCS with a complete recovery and a symptom-free follow-up of 12 months.

Surgical interventions aimed at removing an elongated styloid process can be conducted through intraoral or extraoral methods. The intraoral approach may incorporate advanced techniques such as endoscopy or robotic surgery [29]. However, this method is not without its limitations, especially in cases where vascular proximity is a concern, depending on the angulation of the styloid. A comparable challenge arises with the innovative Minimally Invasive Cervical Styloidectomy (MISC) approach [27]. In instances where the elongated styloid process poses a risk of compressing the internal carotid artery (ICA), it becomes imperative to establish a surgical field exposure through extensive neck dissection. This step is crucial for ensuring a comprehensive approach to the potential vascular complications associated with the elongated styloid process. The choice of the surgical technique, whether intraoral, extraoral, or utilizing novel methods such as MISC, requires careful consideration based on the specific anatomical factors and potential risks involved in each case.

Elongated styloid processes have the potential to elicit a spectrum of symptoms through compression or irritation of neighboring structures, encompassing nerves (IX, X, sympathetic trunk), blood vessels (internal jugular vein and carotid arteries), as well as adjacent soft tissues. Typically, patients manifest discernible symptoms such as unilateral neck and orofacial pain, commonly referred to as stylalgia, accompanied by a peculiar sensation of a foreign body in the throat. This characteristic presentation often facilitates a relatively straightforward diagnosis.

Nonetheless, establishing a conclusive association between the elongated styloid process and conditions such as stroke requires additional thorough assessment and a meticulous correlation of clinical findings. Unraveling the intricate relationship between the elongated styloid process and potential vascular complications demands a comprehensive medical examination considering the multifaceted interplay of anatomical factors and the diverse symptomatic manifestations observed in affected individuals.

## 5. Conclusions

In the culmination of this study, it is imperative to recognize the styloid process as a potential risk factor in the diagnosis of carotid artery dissection (CAD) among young individuals, a factor often overlooked in clinical assessments. After diligently excluding other potential causes, the styloid process emerges as a noteworthy consideration in understanding the complex landscape of CAD etiology. Styloidectomy, as demonstrated in this research, stands out as an efficient and viable technique for effectively managing disturbances in blood circulation associated with the compressed and dissected artery.

To conduct a thorough and comprehensive assessment of potential anatomical factors contributing to CAD, it is recommended that the differential diagnosis incorporate the utilization of angioCT. This advanced imaging technique enables a nuanced exploration of vascular structures, providing valuable insights into the intricate details of carotid artery dynamics and aiding in the accurate identification and management of CAD in young individuals.

## Figures and Tables

**Figure 1 diseases-12-00024-f001:**
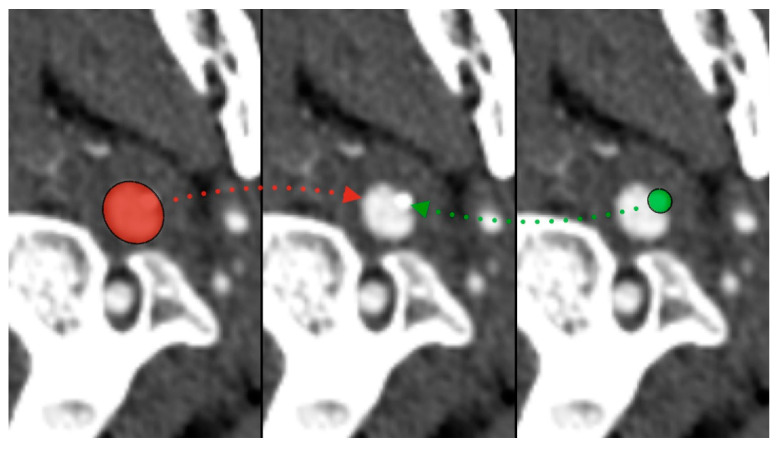
Styloid process compressing left internal carotid artery—red (ICA), green (styloid).

**Figure 2 diseases-12-00024-f002:**
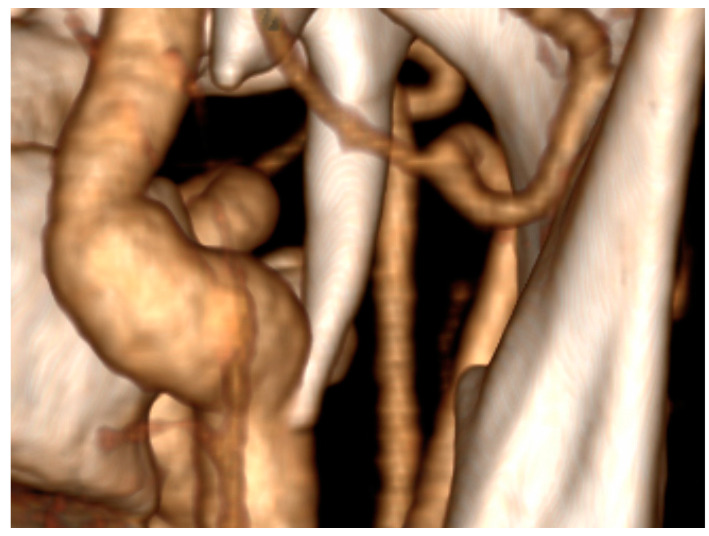
Styloid process compressing left internal carotid artery—3D reconstruction.

**Figure 3 diseases-12-00024-f003:**
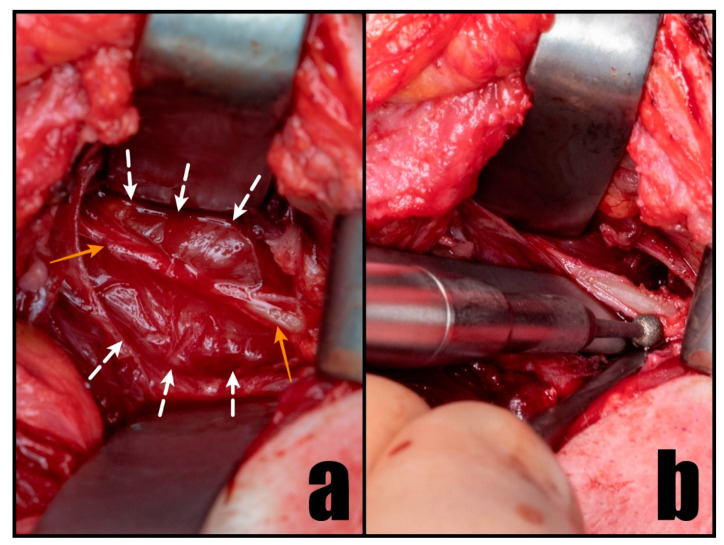
Intraoperative perspective illustrating the compression on LICA by the stylohyoid chain (**a**) and its careful removal using a diamond burr (**b**). The white arrows designate the LICA, while the orange arrows highlight the stylohyoid chain.

**Figure 4 diseases-12-00024-f004:**
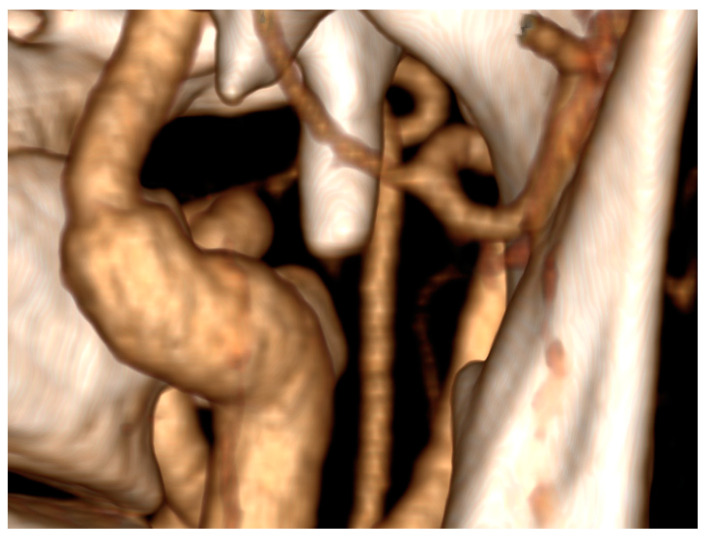
Lack of compression on left internal carotid artery (LICA) by the styloid process after surgery.

## Data Availability

Restrictions apply to the availability of these data. Data were obtained from patients treated at the Department of Cranio-Maxillofacial Surgery, Cracow, Poland, and cannot be shared, in accordance with the General Data Protection Regulation (EU) 2016/679.

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
