# Peer review of "Stylohyoid Chain Syndrome (Eagle Syndrome) in Conjunction with Carotid Artery Dissection: A Case Report of Surgical Treatment"

_diseases, 2024, doi:10.3390/diseases12010024_

Round 1
Reviewer 1 Report
Comments and Suggestions for Authors
I did not know this syndrome until now, thus I read the manuscript reporting a rare clinical case of acute dissection of internal carotid artery due to external compression of abnormally elongated styloid process with interest. Clinical presentation and discussion are properly presented, and illustrations are very useful to better understand the anatomical interactions between bone and vascular structures. I have minor comments to propose:
#1. Given its rarity, which is the estimated incidence (and prevalence) of the syndrome in the general population?
#2. Are there any specific gender or ethnic group prone to developing this syndrome?
#3. Is there a preferred laterality? In the present case, left internal carotid artery is involved.
Comments on the Quality of English Languageminor editing would ameliorate the text
Author Response
Dear Reviewer,
Thank you for your insightful comments and questions regarding our manuscript on a rare acute dissection of the internal carotid artery due to external compression by an abnormally elongated styloid process. We are pleased that you found the clinical presentation, discussion, and illustrations helpful in understanding the complex anatomical interactions.
Responding to your queries:
-
We appreciate you pointing out the need for incidence data in our manuscript. The estimated incidence and prevalence of Eagle syndrome in the general population have now been included in the revised manuscript.
-
Regarding the predilection for any specific gender or ethnic group, Eagle syndrome does show a higher occurrence in women, while carotid artery dissection tends to be more common in men. However, there is no known ethnic predisposition to this vascular syndrome as ethnicity does not influence the pathophysiological mechanisms.
-
As for laterality, it is challenging to determine a preferred side of occurrence due to the scant number of cases reported in the literature. The left internal carotid artery was involved in the current case, but this does not suggest a pattern given the limited data available.
We trust these responses address your concerns and have amended the manuscript accordingly to reflect these points more clearly.

Reviewer 2 Report
Comments and Suggestions for Authors
The authors present a case report based on the effective removal of an elongated styloid process compressing the carotid artery and apparently causing artery dissection. After a first endovascular step, the authors decided to perform styloidectomy to avoid the use of stenting, due to a clear radiological evidence of direct compression. The novelty of the paper, that is extremely too long for a case report and describes really nothing new, would be based on the fact that they performed an emergency styoloidectomy to decompress the artery. Nonetheless, this was not an emergency surgery, thrombectomy was and it was resolutive on symptoms. The fact that the authors claim this was performed without previous stenting of the carotid is definitely intereesting, because stenting might increase the chances of carotid perforation and adds risks due to more manipulation of the vessel endovascularly. however, although very well and exhaustively described, this paper might have been more fit for a short technical note. The whole discussion about the causes of carotid artery dissection has nothing to do with the case in object and although the so called carotid artery syndrome may be one of the causes to be investigated in spontaneous artery dissection, it remains a rare event. I suggest the authors to re-edit the paper in a shorter and more focused version, simply stating that surgery is potentially resolutive in this rare condition, not requiring stents and the ensuing anticoagulation
Author Response
Dear Reviewer,
Thank you for your valuable feedback. We acknowledge your concerns regarding the length and content of our case report. We wish to highlight that the detailed description was partly due to the journal's requirement for a minimum word count of 2500, which necessitated a thorough exploration of the case.
The term 'emergency' in our report referred specifically to the urgent need for decompression due to direct artery compression, rather than the underlying cause, which we believe was accurately depicted in the publication.
Our conclusions have emphasized that surgery is potentially resolutive for this rare condition.
Reviewer 3 Report
Comments and Suggestions for Authors
The authors introduced internal carotid artery dissection due to Eagle syndrome and styloidectomy was successfully done with subsequent 12-month-symptom-free period.
However, it is well known that internal carotid artery dissection is occasionally induced by Eagle syndrome and the treatment. Therefore, the case report is unfortunately scientifically clinically novel.
Author Response
Dear Reviewer,
We appreciate your critical evaluation of our case report and recognize the well-established correlation between Eagle syndrome and internal carotid artery dissection. However, we would like to point out that our publication is the first to document a surgical approach to the decompression of a dissected carotid artery in Eagle syndrome without the use of a stent shortly after a vascular incident.
This specific approach, favoring rapid intervention without stenting, followed by a significant symptom-free period, adds a unique contribution to the existing literature. This detailed documentation of the patient's progress and the decision-making process involved in avoiding stent placement provides valuable clinical insights that could influence future management of similar cases.
We hope this clarification underscores our case report's novelty within the scientific and clinical community.
Round 2
Reviewer 2 Report
Comments and Suggestions for Authors
No significant changes when compared to the previous version, only a revision of english language. The paper remains a case report extrapolated from a previously published series and it is definietely too long to be even a case report, it should be limited to a technical note.
Author Response
Thank you for your feedback. I want to clarify that I received clear instructions from the Editor to increase the word limit to 2500 words, which aligns with the journal's guidelines for Case Reports. This was done in accordance with the editorial requirements, and the paper has been revised to meet these specifications. If you have any further suggestions or concerns, please feel free to let me know, and I will address them accordingly.
Reviewer 3 Report
Comments and Suggestions for Authors
Thank you for the explanation to make it clearer. The authors should mention that the surgery did not need stenting.
Author Response
Thank you very much for your valuable feedback. While there is information in the Results section indicating that there was no need for stent grafting, I have included the crucial detail that vascular surgeons recommended LICA decompression without stenting to make this point clearer. Your suggestion has been incorporated into the revised manuscript.